# Stability and Extraction of Vanillin and Coumarin under Subcritical Water Conditions

**DOI:** 10.3390/molecules25051061

**Published:** 2020-02-27

**Authors:** Ninad Doctor, Grayson Parker, Katie Vang, Melanie Smith, Berkant Kayan, Yu Yang

**Affiliations:** 1Department of Chemistry, East Carolina University, Greenville, NC 27858, USA; doctorn11@students.ecu.edu (N.D.); Parkerg12@students.ecu.edu (G.P.); vangk13@students.ecu.edu (K.V.); smithm15@students.ecu.edu (M.S.); 2Department of Chemistry, Arts and Sciences Faculty, Aksaray University, 68100 Aksaray, Turkey; BerkantKayan@hotmail.com

**Keywords:** vanillin, coumarin, subcritical water extraction, SBWE, vanilla beans, tonka beans, stability, HPLC

## Abstract

In order to facilitate the development of the green subcritical water chromatography technique for vanillin and coumarin, the stability of the compounds under subcritical water conditions was investigated in this work. In addition, their extraction from natural products was also studied. The stability experiments were carried out by heating the mixtures of vanillin and water or coumarin and water at temperatures ranging from 100 °C to 250 °C, while subcritical water extractions (SBWE) of both analytes from vanilla beans and whole tonka beans were conducted at 100 °C to 200 °C. Analyte quantification for both stability and extraction studies was carried out by HPLC. After heating for 60 min, vanillin was found to be stable in water at temperatures up to 250 °C. While coumarin is also stable at lower temperatures such as 100 °C and 150 °C, it undergoes partial degradation after heating for 60 min at 200 °C and higher. The results of this stability study support green subcritical water chromatographic separation and extraction of vanillin and coumarin at temperatures up to 150 °C. The SBWE results revealed that the extraction efficiency of both analytes from vanilla beans and tonka beans is significantly improved with increasing temperature.

## 1. Introduction

As one of the most important aromatic flavor compounds, vanillin (4-hydroxy-3-methoxybenzaldehyde; Figure 1) is massively produced through chemical synthesis, topping ten thousand tons per year worldwide [1]. It is widely used as a food additive and flavor enhancer in beverages, ice-cream, cookies, chocolate, and pudding [2]. Due to its phenolic character, vanillin is also used in the production of pharmaceuticals and fine chemicals [3].

Coumarin (1,2-benzopyrone; Figure 1) is the parent compound of a large class of naturally occurring phenolic constituents [4]. It can be found in many plants, such as lavender, licorice, strawberries, apricots, cherries, cinnamon, and sweet clover [5]. It has a variety of biological activities and a wide range of therapeutic properties [6]. Coumarin has a sweet herbaceous odor and has been detected in some vanilla products as well as in cinnamon, in addition to many other plants. Due to its strong sweet herbaceous odor, coumarin is a popular ingredient in fragrances [4]. However, researchers suspected that coumarin has a genotoxic and carcinogenic effect [7]. Today, coumarin is discontinued or restricted in food flavoring due to its toxicological concerns [7].

Currently, high-performance liquid chromatography (HPLC) is the most technique most used to analyze vanillin and coumarin [4,7,8]. Although HPLC is a popular separation and analysis technique, it requires harmful organic solvents. Subcritical water chromatography is a green alternative to HPLC. The term ‘subcritical water’ refers to water that is heated and pressurized at conditions below its critical point [9,10,11,12]. Subcritical water acts like a weak polar organic solvent when heated [9,10,11,12]. When subcritical water is used as the mobile phase to achieve chromatographic separation, the process is termed as subcritical water chromatography, and has been found to be a very promising technique [12,13,14,15,16,17,18,19,20,21,22,23]. Advantages of subcritical water chromatography include the elimination of hazardous organic solvents required to separate solutes and its relatively fast analysis time [9,10,11,12]. Recent publications demonstrate a great potential for separation of vanillin and coumarin using subcritical water chromatography [15,19]. As shown in our previous work, vanillin and coumarin were successfully separated using the green subcritical water chromatography technique with only water as the mobile phase on different columns [19]. In order to further evaluate the feasibility of employing green subcritical water chromatography for separation and analysis of vanillin and coumarin, as well as the subcritical water extraction of the two analytes from beans, the stability of the analytes in subcritical water needs to be investigated due to potential analyte degradation [24,25,26].

Vanillin partition coefficient in water-supercritical CO_2_ and supercritical carbon dioxide extraction of aqueous vanillin solution containing salts have been recently investigated [27,28]. Several other papers have reported the extraction of vanillin from vanilla beans and coumarin from tonka beans by using organic solvents such as methanol, ethanol, acetonitrile, acetone, chloroform, and hexane [29,30]. However, the organic solvents used in these extractions are toxic and expensive [31].

In this research, the stability of vanillin and coumarin in water at varying temperatures was studied. A mixture of the analyte and water was heated at a predetermined temperature for 15 and 60 min. Spiked studies on SBWE of vanillin and coumarin were then conducted. Finally, vanillin and coumarin were extracted from vanilla beans and tonka beans, respectively, using a homemade subcritical water extraction system at 100 to 200 °C. The heated vanillin-water and coumarin-water mixtures for the stability study and the SBWE extracts were analyzed by HPLC for quantification of vanillin and coumarin.

## 2. Results and Discussions

### 2.1. Stability of Vanillin and Coumarin in Subcritical Water

Figure 2 shows the chromatogram of a standard solution, while Figure 3 demonstrates the chromatograms of vanillin obtained at four different temperatures.

In Figure 3, no extra peaks are observed, indicating that no degradation products are found. The percent recovery of analytes was calculated using the mass found after heating, divided by the mass added before the heating. The mass found after the heating was determined by HPLC analyses. As shown in Table 1, the vanillin recovery after heating for 15 and 60 min remains almost unchanged. The recovery of vanillin after heating for 60 min ranges from 93% at 100 °C to 98% at both 150 and 200 °C, as shown in Table 1. Thus, no significant degradation of vanillin occurred under the subcritical water conditions investigated. Kawamura et al. reported that the subcritical water extraction yield for vanillin from oil palm trunk continued to increase between 100 °C to 200 °C [32]. This may suggest that no degradation of vanillin occurs, and the compound is stable in water at temperatures up to 200 °C during the process of subcritical water extraction. Thus, our vanillin stability findings agree with that reported by Kawamura et al.

The chromatograms of coumarin obtained at various temperatures are shown in Figure 4. There are no extra peaks observed, indicating that no degradation products are found at 278 nm. The quantitative stability results are given in Table 2. It should be pointed out that no coumarin degradation was found at any of the temperatures tested after heating for 15 min, as shown in Table 2. However, the percent recovery of coumarin decreased to 91% and 82% at 200 °C and 250 °C, respectively, after heating for 60 min, as shown in Table 2. This means that there is a minor degradation of coumarin at 200 °C and 250 °C. Although Kawamura et al. did not extract coumarin in their work, they did report that the subcritical water extraction yield for p-coumaric acid reached a maximum at 160 °C and then decreased at 180 °C and 200 °C [32]. This result suggests that p-coumaric acid may be degraded at temperatures higher than 160 °C. This is in agreement with our findings for coumarin’s stability in subcritical water.

The stability results for coumarin and vanillin are encouraging. Because both solutes are stable at temperatures up to 150 °C, green separation techniques for coumarin and vanillin, such as subcritical water chromatography and subcritical water extraction, can be developed and safely employed. Since coumarin is widely found in many plants such as lavender, licorice, strawberries, apricots, cherries, cinnamon, and sweet clover [5], the green approach of subcritical water extraction of coumarin from natural products is of great interest. The traditional way of extracting coumarin from natural products involves hazardous organic solvents. Using subcritical water extraction should make the process greener.

### 2.2. Subcritical Water Extraction

The SBWE recovery of vanillin and coumarin from the spiked samples is given in Table 3. Although the recovery for both analytes achieved at 100 °C is slightly lower than that obtained at 150 and 200 °C, the recoveries are around 100% for extractions at all three temperatures. The relative standard deviations are within 1%. The reason for the high extraction efficiency is most likely due to the lack of the matrix effect in the spiked sample.

Figure 5 shows the chromatogram of vanilla beans after subcritical water extraction at 100 °C. Since there is no coumarin in vanilla beans, it was used as the internal standard for separation and determination of vanillin. The chromatogram of tonka beans after subcritical water extraction at 100 °C is shown in Figure 6. As one can see from Figure 5 and Figure 6, both vanillin and coumarin were well separated as major peaks in beans samples. The results of subcritical water extraction of vanillin and coumarin are given in Table 4. It should be noted that as temperature was raised from 100 °C to 150 °C and 200 °C, the extracted quantity of vanillin and coumarin was significantly increased. This confirms that there is a matrix effect on subcritical water extraction efficiency in real world samples.

In order to further evaluate SBWE efficiency, we carried out additional extraction of vanillin from vanilla beans at 150 °C for 8 h. The vanillin quantity obtained by SBWE under this condition is 47.4 mg/g, which is very close to the 49.7 mg/g achieved by sonication extraction.

## 3. Materials and Methods

### 3.1. Reagents and Materials

Vanillin and coumarin were purchased from ACROS Organics (New Jersey, USA). HPLC grade methanol and phenol were obtained from Fischer Scientific (Fair Lawn, NJ, USA). Acetone was acquired from VWR International (West Chester, PA, USA). The 18 MΩ-cm deionized water was prepared in the laboratory using a PureLab ultra MK2 system from ELGA (Lane End, Buckinghamshire, England). Adsorbosil C18 (4.6 × 150 mm, 5 µm) was bought from Alltech Associates, Inc. (Deerfield, IL, USA). Stainless steel vessels (7.07 mL, 9 cm × 1 cm ID) were purchased from Raleigh Valve and Fitting Company (Raleigh, NC, USA). Caffeine was acquired from Eastman Kodak Co. (Rochester, NY, USA). Vanilla beans were purchased from Florida Herb House (Daytona, FL, USA), while whole tonka beans were bought from Amazon.

### 3.2. Heating of Organic-Water Mixtures

The 7.07 mL stainless steel reaction vessels were used to heat the vanillin-water or coumarin-water mixtures. Prior to use, the vessels were rinsed with acetone and allowed to dry completely. Two layers of Teflon tape were wrapped around both ends on the vessel and one end was tightly sealed with an end cap. Approximately 5 mg of vanillin or coumarin was accurately weighed into each reaction vessel. Then, 3 mL of deionized water was added to each reaction vessel. For safety considerations, the remaining volume inside the vessel was left void. The other cap was placed on the vessel and tightened with a vise to insure a complete seal.

The loaded vessels were placed onto a metal tray, which was then put into a gas chromatograph oven. The mixtures were heated at 100, 150, 200, or 250 °C, each for 60 min. The heating time was started once the oven reached the set temperatures. Please note that there is a temperature lag time between the vessels’ temperature and the oven’s display temperature, and it is 5, 8, 12, and 15 min, for 100, 150, 200, and 250 °C, respectively. A thermocouple was used to monitor the temperature of the vessels. The counting of the heating time excluded the temperature lag time to ensure that the sample temperature reached the intended test temperature.

After heating, the vessels were removed from the oven and allowed to cool to room temperature. The vessels were then set standing and one end cap was removed. The vanillin-water or coumarin-water mixture in a given vessel was removed into a 10 mL glass vial. The vessel and the end cap were rinsed with 0.5 mL of methanol to remove residual analyte. The rinse was then added to the glass vial containing the mixture. A second rinse was repeated with another 0.5 mL of methanol and the rinse solution was again combined into the 10 mL collection vial. An internal standard, phenol, was added to the vial prior to HPLC analysis. The volume of the sample was adjusted before HPLC analysis to ensure that the absorbance of the sample was within the linear range of the calibration curve.

### 3.3. Preparation of Standard Solutions

The standard stock solutions were made by weighing out approximately 100 mg of coumarin or vanillin into two separate 25 mL volumetric flasks. Methanol was added to each flask to the mark. Each internal standard stock solution was prepared by weighing a given amount of an appropriate internal standard (phenol for stability study, coumarin for SBWE of vanillin, or caffeine for SBWE of coumarin) into a 25 mL volumetric flask and methanol was added to the mark. These stock solutions were used to prepare three standard solutions to generate an HPLC calibration curve. The concentration of the HPLC calibration solutions ranges from 50 to 500 ppm. Thus, the linear range for this HPLC method is at least two orders of magnitude. Because the focus of this study is to evaluate analyte stability and subcritical water extraction, not to develop a chromatographic method, we did not use concentrations beyond this range. Therefore, the linear range of this HPLC method may be much wider.

### 3.4. Sonication Extraction

Extraction of vanillin and coumarin was performed by using a sonicator. At the beginning, 1.000 g of vanillin and coumarin crushed samples were placed inside separate 7 mL glass vials. Then, 5 mL of methanol was added. These vials were then sealed with aluminum lines caps and parafilm. They were then placed inside a beaker with water and placed inside the sonicator. The sonication extraction was performed for 30 min at room temperature. After 30 min, each sample was filtered through a Whatman GDX filter into a glass vial and 30 µL of the internal standard solution was added to each vial. Triplicate sonication extractions were conducted.

### 3.5. Subcritical Water Extraction

The SBWE recovery of vanillin and coumarin was evaluated at 100–200 °C. A 7.07 mL stainless steel vessel was used for the recovery studies. One end was connected to the vessel tube, which was filled with clean sand until full. A piece of Kimwipes spiked with the two analytes (100 µg each) was then placed inside the extraction vessel. Then, the extraction vessel was placed in a gas chromatograph. An ISCO model 260 D syringe pump was used to supply 18 MΩ-cm water. Leak detection was carried out at 50 atm. A preselected temperature was then set and the oven was turned on. Note that there is a time delay between the actual and displayed temperatures. This delay was 12 min for 100 °C, 16 min for 150 °C, and 18 min for 200 °C. After 30 min static extraction, 3 mL of extractant was collected into a 7 mL collection vial. Immediately after the collection, 1 mL of methanol was added to the vial.

To investigate the subcritical water extraction of vanillin and coumarin from vanilla beans and whole tonka beans, vanilla beans and tonka beans were cut into small and uniform pieces. Approximately 1 g of cut vanilla beans or tonka beans were loaded into the extraction vessel. To reduce void volume, clean sand was packed into the vessel until it was full. The SBWE was conducted at 100 °C, 150 °C, and 200 °C in the same manner as the recovery studies described above.

### 3.6. HPLC Analysis

Shimadzu Nexera UFLC system (Shimadzu Corporation, Chiyoda-ku Tokyo, Japan) was used for quantification of vanillin and coumarin in the stability study. The separation was carried out on an Adsorbosil C18 column. The mobile phase consisted of 70% water and 30% methanol. The flow rate was set at 1 mL/min and injection volume was 10 μL. The UV detector was set to 278 nm for solute detection. It should be pointed out that a green subcritical water chromatography method could be used here. However, the standard HPLC method was employed in this work to ensure the accuracy of the stability evaluation.

For quantification of vanillin and coumarin after subcritical water extraction of vanilla beans and tonka beans, the same Adsrobosil C18 column was used. The mobile phase consisted of 60% water and 40% methanol. The other HPLC conditions are the same as the HPLC analysis for the stability studies.

## 4. Conclusions

In summary, after 15 and 60 min of heating, vanillin remained relatively stable in water at all four temperatures tested—100, 150, 200, and 250 °C. However, longer heating time causes greater degradation of coumarin at higher temperatures. Although after 60 min of heating, coumarin was still stable at 100 and 150 °C, a slight degradation of coumarin occurred at 200 and 250 °C. Based on the findings of this study, subcritical water can be safely used for extraction of vanillin and coumarin at temperatures up to 150 °C because both analytes are clearly stable in subcritical water at or below 150 °C. While high subcritical water extraction recoveries have been achieved for extractions of vanillin and coumarin from spiked samples at all three temperature investigated, the subcritical water extraction efficiency for vanillin and coumarin from vanilla and tonka beans increased with higher extraction temperature. SBWE of vanillin from vanilla beans at 150 °C for 8 h yielded comparable vanillin quantity as that obtained by sonication extraction.

## Figures and Tables

**Figure 1 molecules-25-01061-f001:**
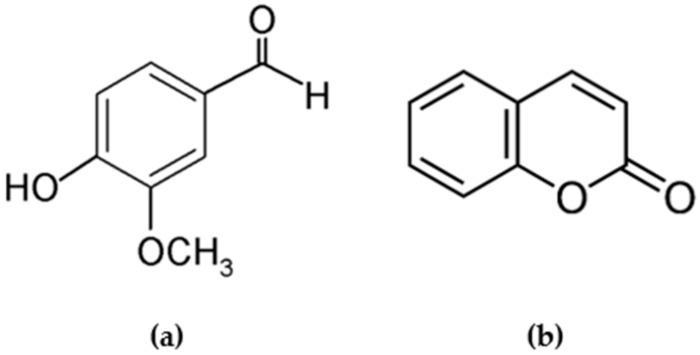
Structure of (**a**) vanillin and (**b**) coumarin.

**Figure 2 molecules-25-01061-f002:**
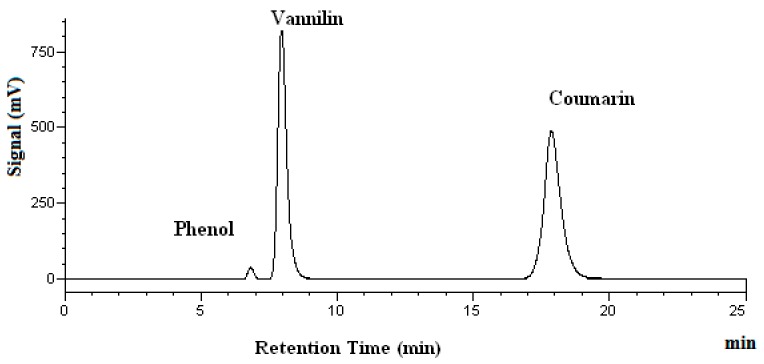
Chromatograms of vanillin and coumarin in a standard solution. Phenol was used as the internal standard.

**Figure 3 molecules-25-01061-f003:**
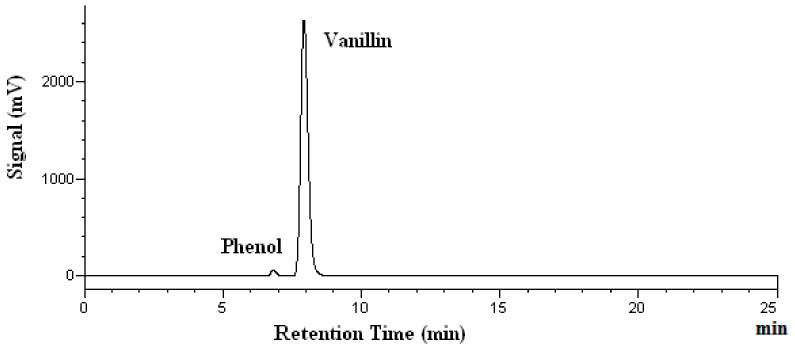
Chromatograms of vanillin after heating at 200 °C. Phenol was used as the internal standard.

**Figure 4 molecules-25-01061-f004:**
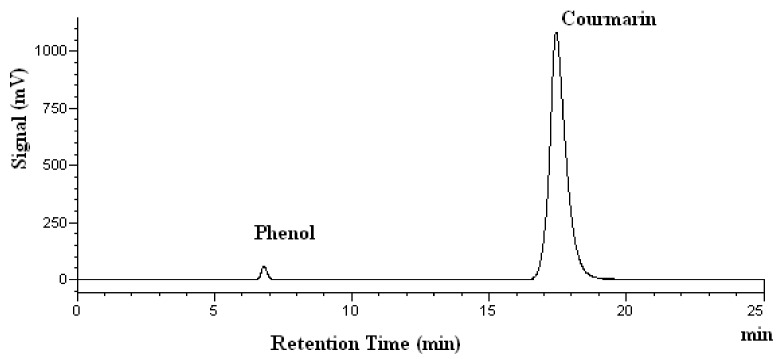
High-performance Liquid Chromatography chromatograms of coumarin after heating at 200 °C. Phenol was used as the internal standard.

**Figure 5 molecules-25-01061-f005:**
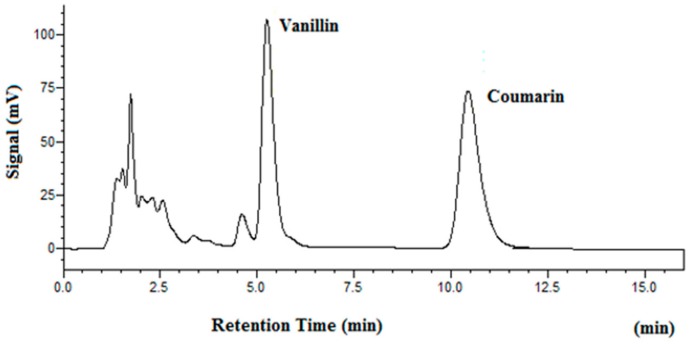
Chromatogram of vanilla beans after subcritical water extraction at 100 °C (internal standard: coumarin).

**Figure 6 molecules-25-01061-f006:**
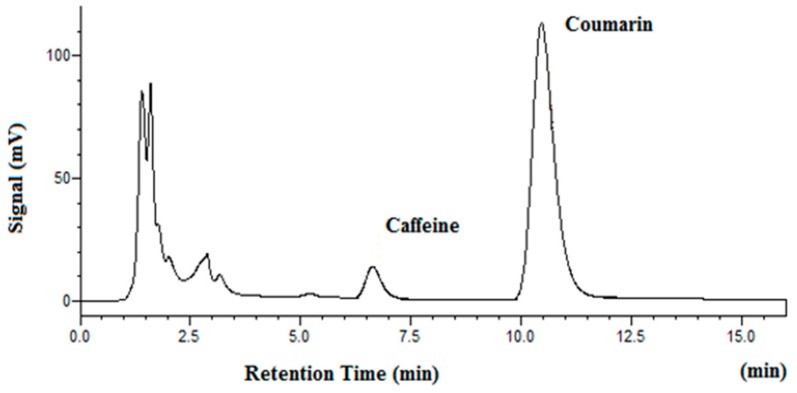
Chromatogram of whole tonka beans after subcritical water extraction at 100 °C (internal standard: caffeine).

**Table 1 molecules-25-01061-t001:** Average percent recovery of vanillin obtained at different temperatures after heating.

Temperature	15 min		60 min	
(°C)	% Recovery	% RSD ^a^	% Recovery	% RSD ^a^
100	97	5	93	6
150	103	5	98	2
200	97	2	99	2
250	99	1	95	2

^a^ Based on triplicate measurements.

**Table 2 molecules-25-01061-t002:** Percent recovery of coumarin obtained at different temperatures after heating.

Temperature	15 min		60 min	
(°C)	% Recovery	% RSD ^a^	% Recovery	% RSD ^a^
100	98	6	96	7
150	101	4	100	7
200	97	3	91	10
250	101	5	82	20

^a^ Based on triplicate measurements.

**Table 3 molecules-25-01061-t003:** Recovery of vanillin and coumarin obtained by 30-min subcritical water extraction of the spiked samples.

Temperature	Vanillin		Coumarin	
	% Recovery	% RSD ^a^	% Recover	% RSD ^a^
100 °C	98.4	0.6	96.9	0.7
150 °C	104.2	0.8	102.3	0.8
200 °C	104	0.6	102.2	0.6

^a^ Triplicate measurements.

**Table 4 molecules-25-01061-t004:** Comparison of analyte quantity from vanilla beans obtained by 30-min subcritical water extraction and sonication extraction.

Temperature (°C)	Vanillin in Vanilla Beans		Coumarin in Tonka Beans	
	mg/g	% RSD ^a^	mg/g	% RSD ^a^
100	1.07	9	23.9	6
150	1.66	3	25.2	3
200	1.95	5	36.8	9
Sonication	4.97	5	102.5	5

^a^ Based on triplicate measurements.

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
