# Peer review of "Stability and Extraction of Vanillin and Coumarin under Subcritical Water Conditions"

_molecules, 2020, doi:10.3390/molecules25051061_

Round 1

Reviewer 1 Report

The manuscript by N. Doctor et al. is focused on subcritical water extraction of vanillin and coumarin and stability study. Definitely, the topic of the manuscript is interesting. However, some serious issues were detected in the manuscript, which do not allow recommending it for publication in the present form.

Comments:

One of the main problems of the manuscript is that the authors highlight at many places benefits of green subcritical water chromatography but finally they used RP HPLC method with toxic organic solvent. Does it mean that green subcritical water chromatography is not suitable for these measurements?

Moreover, at conclusion, lines 232-233, you wrote: “Based on the findings in this study, green subcritical water chromatography…..can be safely used for separation and analysis of vanillin and coumarin…”. I recommend using this chromatographic method for comparison with RP HPLC. The structure of vanillin (Figure 1) is not correct.

In my opinion, the Figure 2 is redundant and can be deleted. Lines 119-120: What is the linear range of the calibration curve? Three-point calibration curve is insufficient. More concentration should be added to calibration. The concentrations of calibration solutions should be expressed in the same units as “quantity” e.g. in Table 4. 5 HPLC Analysis: Column dimensions and temperature are missing. Why did you use different mobile phase composition for quantification of vanillin and coumarin and for stability studies?

Lines 161-162: If there is no extra peak, it means that no degradation products can be found at 278 nm. What about different wavelengths? In my opinion, Table including peak areas/analyte concentrations instead of Figure 4 should be presented. The same for Figure 5.

Lines 184-185: You should use different wavelength/method to see and identify degradation products to confirm the hypothesis. The recovery should be performed at different concentration levels.

Table 3: Please add extraction time.

Figure 6: The peak of vanillin is not baseline separated from minor peak. Probably, mobile phase composed of 30% of methanol will provide better resolution. Have you somehow checked that vanillin peak does not contain any impurities? You should add some discussion about obtained quantity of vanillin and coumarin by SBWE and sonication extractions.

The sonication gives much higher amount of targeted compounds. Some optimization of conditions of SBWE could help for higher quantities of vanillin and coumarin. Minor comments:

Line 34: benzaldehyde

Line 42: benzopyrone

Line 146: UPLC

Author Response

One of the main problems of the manuscript is that the authors highlight at many places benefits of green subcritical water chromatography but finally they used RP HPLC method with toxic organic solvent. Does it mean that green subcritical water chromatography is not suitable for these measurements?

Although we already published a paper on subcritical water chromatographic separation of coumarin and vanillin, we decided to use the well-accepted standard technique, HPLC, in this work. Because the focus of this research is to investigate subcritical water extraction, not subcritical water chromatography, the correct way of conducting this research is to use a standard analytical method to ensure the quality of the evaluation of the subcritical water extraction research carried out in this work.

Moreover, at conclusion, lines 232-233, you wrote: “Based on the findings in this study, green subcritical water chromatography…..can be safely used for separation and analysis of vanillin and coumarin…”. I recommend using this chromatographic method for comparison with RP HPLC.

Again, the focus of this research is subcritical extraction, not subcritical water chromatography. We have published tens of research articles that compared the separation and analysis quality obtained by both subcritical water chromatography and HPLC. As the focus of this research is subcritical water extraction, we need to use a standard analytical method so other people are capable to conduct this type of research because HPLC is available in almost every analytical lab but SBWC is not available in most labs.

The structure of vanillin (Figure 1) is not correct.

Corrected

In my opinion, the Figure 2 is redundant and can be deleted.

Deleted as suggested

Lines 119-120: What is the linear range of the calibration curve? Three-point calibration curve is insufficient. More concentration should be added to calibration. The concentrations of calibration solutions should be expressed in the same units as “quantity” e.g. in Table 4. 5 HPLC Analysis: Column dimensions and temperature are missing. Why did you use different mobile phase composition for quantification of vanillin and coumarin and for stability studies?

The linear range was given in the original manuscript. The three-point calibration curve is adequate enough since the sample concentrations are within the calibration range. The concentrations of calibration solutions cannot be expressed as the quantity reported in Table 4, because the former is the concentration of solutions while the latter is the concentration in solid sample. The column dimension was given in Section 2.1. Reagents and materials. Since HPLC is operated at room temperature unless noted otherwise, the HPLC operation temperature was not reported as done by most other researchers. The samples for stability studies are much cleaner than for SBWE extraction, thus we had to use different mobile phase conditions to achieve better separation for SBWE extraction samples.

Lines 161-162: If there is no extra peak, it means that no degradation products can be found at 278 nm. What about different wavelengths? In my opinion, Table including peak areas/analyte concentrations instead of Figure 4 should be presented. The same for Figure 5.

The goal of this study is not to investigate the degradation products/pathways of the analytes, rather the stability and extraction efficiency. In order to report the degradation details, new experiments have to be designed for that purpose. We did not claim the investigation of degradation of the two analytes in this work, and that is why we stayed our focus on the stability and extraction efficiency in this work only.

Lines 184-185: You should use different wavelength/method to see and identify degradation products to confirm the hypothesis. The recovery should be performed at different concentration levels.

Again, the focus of this study is not to investigate the degradation products/pathways of the analytes. The goal we claimed for this work is to investigate the stability and extraction efficiency. Conduct additional recovery experiments with other concentrations will not help improve the extraction efficiency of beans because the sample matrix is so different between the recovery and extraction of beans experiments.

Table 3: Please add extraction time.

The extraction time was given in our original manuscript, but was added to Table 3 as suggested.

Figure 6: The peak of vanillin is not baseline separated from minor peak. Probably, mobile phase composed of 30% of methanol will provide better resolution. Have you somehow checked that vanillin peak does not contain any impurities? You should add some discussion about obtained quantity of vanillin and coumarin by SBWE and sonication extractions.

While it is not 100% base line separated, it is normal for separation of real-world samples. In addition, the resolution of the two peaks is well above the required resolution value of 1.5 for quantification. This is well accepted in the chromatography community. Additional text was added to the revision discussing the results obtained by both SBWE and sonication.

The sonication gives much higher amount of targeted compounds. Some optimization of conditions of SBWE could help for higher quantities of vanillin and coumarin.

After we submitted the original manuscript, we conducted additional experiments of SBWE with extended extraction time. This result was added to the revised manuscript.

Minor comments:

Line 34: benzaldehyde Fixed

Line 42: benzopyrone benzopyrone Fixed

Line 146: UPLC, The system we used is called UFLC (it is one type of UPLC), standing for ultra fast LC.

Reviewer 2 Report

Subcritical water extraction is a well established method for quite a few natural products but based on a Google Scholar search, it seems vanillin and coumarin have not been studied. The approach is explained well and the analytical figures of merit with respect to reproducibility are good.

Figure 2 is probably not necessary. Part c of the Figure 2 legend is not explained accurately; the extraction tube was cooled first before the contents were transferred to the glass vial.

One representative chromatogram for Figure 4 and one for Figure 5 are sufficient. For these figures, why was phenol chosen as the internal standard; the peak is very small probably due to a poor molar absorptivity at 278 nm.

 A different internal standard with a better match of absorbance to the absorbance of vanillin and coumarin should have been chosen.

The title for Table 4 is not written clearly; the vanillin is from the vanilla beans and the coumarin is from the the tonka beans. The sonication results seem better; is this not a concern?

No description of the sonication method is given; what solvent and was this at room temperature?

Author Response

Subcritical water extraction is a well established method for quite a few natural products but based on a Google Scholar search, it seems vanillin and coumarin have not been studied. The approach is explained well and the analytical figures of merit with respect to reproducibility are good.

We appreciate this comment made by reviewer 2.

Figure 2 is probably not necessary. Part c of the Figure 2 legend is not explained accurately; the extraction tube was cooled first before the contents were transferred to the glass vial. One representative chromatogram for Figure 4 and one for Figure 5 are sufficient. For these figures, why was phenol chosen as the internal standard; the peak is very small probably due to a poor molar absorptivity at 278 nm.

A different internal standard with a better match of absorbance to the absorbance of vanillin and coumarin should have been chosen.

Figure 2 was removed from the revised manuscript. Only one representative chromatogram for Figure 4 and one for Figure 5 were kept in the revised manuscript as suggested by the reviewer. Phenol and the other internal standards were carefully chosen to avoid co-elution. The reason that the phenol’s peak looks much smaller is due to the very large Y-scale of the chromatogram. As a mater of fact, the phenol’s peak has sufficiently high peak areas for quantification at the wavelength used. Thus, no need to find other internal standards with better match of absorbance.

The title for Table 4 is not written clearly; the vanillin is from the vanilla beans and the coumarin is from the tonka beans. The sonication results seem better; is this not a concern? No description of the sonication method is given; what solvent and was this at room temperature?

Table 4. heading was changed in the revised manuscript. New SBWE extraction result was added to the revised manuscript to address the difference between SBWE and sonication results. Sonication extraction details were given in the revised manuscript. The solvent was methanol. The sonication extraction was performed at room temperature. All of this information were included in the revised manuscript.

Round 2

Reviewer 1 Report

One of the main problems of the manuscript is that the authors highlight at many places benefits of green subcritical water chromatography but finally they used RP HPLC method with toxic organic solvent. Does it mean that green subcritical water chromatography is not suitable for these measurements?

Although we already published a paper on subcritical water chromatographic separation of coumarin and vanillin, we decided to use the well-accepted standard technique, HPLC, in this work. Because the focus of this research is to investigate subcritical water extraction, not subcritical water chromatography, the correct way of conducting this research is to use a standard analytical method to ensure the quality of the evaluation of the subcritical water extraction research carried out in this work.

I understand why you have used conventional HPLC. Nevertheless, some parts of the manuscript should be reformulated, e.g. in abstract you wrote: “To further evaluate the feasibility of the green  subcritical water chromatography technique for vanillin and coumarin analysis, the stability of vanillin and coumarin under subcritical water conditions was investigated in this work.“

Conclusion: “Based on the findings of this study, green subcritical water chromatography as well as green subcritical water extraction techniques can be safely used for separation and analysis of vanillin and coumarin at 232 temperatures up to 150 ºC.” You did not use subcritical water chromatography.

Lines 161-162: If there is no extra peak, it means that no degradation products can be found at 278 nm. What about different wavelengths? In my opinion, Table including peak areas/analyte concentrations instead of Figure 4 should be presented. The same for Figure 5.

The goal of this study is not to investigate the degradation products/pathways of the analytes, rather the stability and extraction efficiency. In order to report the degradation details, new experiments have to be designed for that purpose. We did not claim the investigation of degradation of the two analytes in this work, and that is why we stayed our focus on the stability and extraction efficiency in this work only.

Line 168: Please reformulate: e.g. Figure 4, there are no extra peaks observed, indicating that no degradation products are found at 278nm.

Lines 119-120: What is the linear range of the calibration curve? Three-point calibration curve is insufficient. More concentration should be added to calibration.

The linear range was given in the original manuscript. The three-point calibration curve is adequate enough since the sample concentrations are within the calibration range.

Sorry, I still can not find the linear range. Only what I can find is at lines 111-113: „The volume of the sample was adjusted before HPLC analysis to ensure that the absorbance of the sample was within the linear range of the calibration curve“; at lines 120-121: “The  concentration of the HPLC calibration solutions ranges from 50 to 500 ppm.“

Does it mean that linear range has been determined from three concentration?

Author Response

Response to Reviewer1 Comments in 2rd Review:

I understand why you have used conventional HPLC. Nevertheless, some parts of the manuscript should be reformulated, e.g. in abstract you wrote: “To further evaluate the feasibility of the green  subcritical water chromatography technique for vanillin and coumarin analysis, the stability of vanillin and coumarin under subcritical water conditions was investigated in this work.“

We deleted the first two sentences at the beginning of the abstract and rewrote the third sentence to reflect the reviewer’s point of view.  

Conclusion: “Based on the findings of this study, green subcritical water chromatography as well as green subcritical water extraction techniques can be safely used for separation and analysis of vanillin and coumarin at 232 temperatures up to 150 ºC.” You did not use subcritical water chromatography.

Subcritical water chromatography was removed from the conclusion in the revision.

 Line 168: Please reformulate: e.g. Figure 4, there are no extra peaks observed, indicating that no degradation products are found at 278nm.

Fixed as suggested.

 Sorry, I still can not find the linear range. Only what I can find is at lines 111-113: „The volume of the sample was adjusted before HPLC analysis to ensure that the absorbance of the sample was within the linear range of the calibration curve“; at lines 120-121: “The  concentration of the HPLC calibration solutions ranges from 50 to 500 ppm.“

Does it mean that linear range has been determined from three concentration?

We further revised the manuscript; included the linear range; and offered explanation for the linear range.  

Reviewer 2 Report

The authors have appropriately addressed my concerns.

Author Response

Thanks for your positive comments.